# Responses of Soil Carbon and Microbial Residues to Degradation in Moso Bamboo Forest

**DOI:** 10.3390/plants13111526

**Published:** 2024-05-31

**Authors:** Shuhan Liu, Xuekun Cheng, Yulong Lv, Yufeng Zhou, Guomo Zhou, Yongjun Shi

**Affiliations:** 1State Key Laboratory of Subtropical Silviculture, Zhejiang A&F University, Lin’an 311300, China; 2Zhejiang Province Key Think Tank, Institute of Ecological Civilization, Zhejiang A&F University, Lin’an 311300, China; 3Key Laboratory of Carbon Cycling in Forest Ecosystems and Carbon Sequestration of Zhejiang Province, Zhejiang A&F University, Lin’an 311300, China; 4School of Environmental and Resources Science, Zhejiang A&F University, Lin’an 311300, China; 5Forestry Bureau of Anji County, An’ji 313300, China

**Keywords:** Moso bamboo forest, degradation, soil microorganism, microbial residual carbon, amino sugars

## Abstract

Moso bamboo (*Phyllostachys heterocycla cv. Pubescens*) is known for its high capacity to sequester atmospheric carbon (C), which has a unique role to play in the fight against global warming. However, due to rising labor costs and falling bamboo prices, many Moso bamboo forests are shifting to an extensive management model without fertilization, resulting in gradual degradation of Moso bamboo forests. However, many Moso bamboo forests are being degraded due to rising labor costs and declining bamboo timber prices. To delineate the effect of degradation on soil microbial carbon sequestration, we instituted a rigorous analysis of Moso bamboo forests subjected to different degradation durations, namely: continuous management (CK), 5 years of degradation (D-5), and 10 years of degradation (D-10). Our inquiry encompassed soil strata at 0–20 cm and 20–40 cm, scrutinizing alterations in soil organic carbon(SOC), water-soluble carbon(WSOC), microbial carbon(MBC)and microbial residues. We discerned a positive correlation between degradation and augmented levels of SOC, WSOC, and MBC across both strata. Furthermore, degradation escalated concentrations of specific soil amino sugars and microbial residues. Intriguingly, extended degradation diminished the proportional contribution of microbial residuals to SOC, implying a possible decline in microbial activity longitudinally. These findings offer a detailed insight into microbial C processes within degraded bamboo ecosystems.

## 1. Introduction

Soil represents the largest reservoir of carbon (C) pool in terrestrial ecosystems, surpassing the amount of vegetation and the atmosphere. As a result, soil serves as a crucial reservoir and sink of C in ecosystems [1]. Contrasting marine domains, terrestrial ecosystems, particularly forests, offer amplified carbon sequestration potential [2]. Forests are particularly significant due to their large and varied biomass, which results in substantial C storage, much of which is held in forest soils. This kind of carbon storage plays a vital role in mitigating the greenhouse effect and global warming [3]. Consequently, exploring forest soil carbon sequestration becomes imperative for achieving carbon equilibrium and mitigating global warming repercussions.

Soil organic carbon (SOC) serves as an essential element in the carbon (C) cycle of ecosystems and is a fundamental component of soil [4,5]. SOC is primarily obtained from a variety of plant litter, decomposed plant bodies, and roots. These materials can provide vital nutrients for Moso bamboo growth and significantly influence soil’s physical and chemical properties [4]. Understanding the role microorganisms play in SOC storage has always posed challenges in soil C pool studies [6]. It is evidenced in various studies that microbial residue carbon (MRC) comprises a large proportion (over 50%) plays an important part of the SOC pool [7,8,9,10]. This finding underscores the crucial role microorganisms play in the formation and turnover of SOC. Thus, the study of microbial residual carbon is a significant aspect of SOC research. One of the form of carbon sequestration is stabilizing microbial residues in Moso bamboo By converting carbon in organic matter into forms that are more resistant to decomposition, microbial processes can effectively “lock away” carbon in the soil for longer periods. Various factors influence the formation and stability of microbial residues, including soil type, moisture, temperature, pH, and the presence of minerals that can bind organic compounds. Agricultural and land management practices, such as tillage, crop rotation, and organic matter additions, can also influence microbial residue formation and stability [7,8,11]. Joergensen [12] discovered that microbial residues primarily consist of chemically stable amino sugars, with glucosamine originating predominantly from fungi, and muramic acid exclusively derived from bacteria. These elements can broadly differentiate the relative contributions of fungal and bacterial residues to SOC. Amino sugars, constituting a part of the soil microorganisms’ cell wall, demonstrate high stability and microbial heterogeneity [13].

These are extensively utilized in the accumulation of soil organic matter, the dynamic alteration of microbial communities, and the carbon and nitrogen cycle. The predominant research methodology involves studying microbial residues through soil amino sugars. Soil microbial residues, present in the form of amino sugars in the soil, are a suitable indicator for evaluating the dynamics of soil microbial residues in different forest types and the contribution of MR to SOC [14,15].

Moso bamboo (*Phyllostachys heterocycla cv. Pubescens*) is a prominent forest type in the subtropical forest ecological system, is pivotal for carbon sequestration, especially significant in countering global warming [16]. Moso bamboo, chiefly found in Southern China, holds ecological, economic, and national forestry importance [17,18,19]. Apart from environmental benefits, its versatility spans from furniture production to being a food source [20,21]. What sets bamboo apart is its short growth cycle of two years, its rapid growth rate, and the short harvest time compared to other forest species, which gives it enormous carbon sink potential in bamboo forest ecosystems [22,23].

The management of Moso bamboo forests has become increasingly challenging due to rising costs and decreasing profits in recent years [14]. Furthermore, the high costs of transportation and exploitation have led to a decline in the intensive management of these forests, resulting in their degradation and abandonment by farmers, particularly in areas with slightly higher elevations or more complex topography [19]. This concerning trend is projected to persist, escalating the abandonment rate of these forests [24]. Song et al. [21] found that intensive management of these forests can increase vegetation carbon stocks but hinder the accumulation of SOC stocks. To comprehend the degradation effects on Moso bamboo ecosystems, we examined variances in their ecosystem soil carbon dynamics across various degradation stages [16,19,23]. The role and changes of soil microbial residues in these degradation contexts remain ambiguous. Our study, pioneering in its application of soil amino sugars and microbial residues in this context, seeks to elucidate shifts in SOC, amino sugar content, and microbial residue contributions under different degradation timelines, using the spatio-temporal substitution approach. The study made the following assumptions: (1) degradation time increases the content of soil SOC, microbial C (MBC), and water-soluble C (WSOC) in Moso bamboo ecosystems; (2) degradation increases the content of soil MR, fungal residue (FR), and bacterial residue (BR), and the effect of degradation on the C content of microbial residues in Moso bamboo forests depends on the length of degradation time; (3) quantify the proportion of MRC in SOC within the Moso bamboo forest. Consequently, this study can provide new and effective strategies for productivity enhancement and restoration of degraded Moso bamboo forests while also improving the understanding of the contribution of microbial residues to soil SOC in Moso bamboo forest ecosystems.

## 2. Results

### 2.1. Changes of Soil Environmental Factors and Soil C Pool under Different Degradation Time

In an examination of soil pH profiles at two distinct depths, there were observable variations between the CK, D-5, and D-10 groups. Within the 0–20 cm depth, the D-5 and D-10 groups showed decrease in pH compared to the CK group, albeit with the difference was not significant (Table 1). This trend was also reflected at the 20–40 cm depth, albeit with the difference was also not significant (Table 2). Regarding soil moisture content (SMC), in the 0–20 cm soil layer, the SMC of CK was significantly higher than that of D-5 and D-10. Moreover, the SMC of D-5 and D-10 decreased by 28.81% and 28.21%, respectively, with the degradation time, which was statistically significant (*p <* 0.05).

The same trend was observed at depths of 20–40 cm, compared to CK, the SMC of D-5 and D-10 decreased by 26.95% and 29.48%, respectively, with the degradation time, which was statistically significant (*p <* 0.05).

The average SOC content of CK, D-5, and D-10 groups at 0–20 cm depth, which were 34.4 ± 5.29, 47.5 ± 8.24, and 74.3 ± 8.07 g·kg^−1^, respectively. The degradation time significantly increased the SOC content in the 0–20 cm soil layer, and the difference was statistically significant (*p <* 0.01) (Figure 1). Similarly, the average SOC content of CK, D-5, and D-10 groups at 20–40 cm depth were 38.9 ± 5.59, 45.5 ± 7.61, and 62.0 ± 7.30 g·kg^−1^, respectively. The degradation time also increased the SOC content in the 20–40 cm soil layer of the Moso bamboo forest. There were significant differences among the three groups (*p <* 0.05). and there were significant differences between CK and D-10 According to the above analysis, degradation years has a significant effect on SOC accumulation, and it can change the soil C pool content. As the degradation time increases, the SOC content continuously accumulates in the soil, resulting in an increase in soil fertility. In summary, degradation year is an essential factor that affects the SOC content in the Moso bamboo forest soil. Therefore, it is necessary to consider the impact of degradation time on SOC when managing and maintaining Moso bamboo forests.

The same trend has been seen in WSOC and MBC: the average WSOC concentrations of CK, D-5, and D-10 at 0–20 cm depth were 193.4 ± 49.0, 262.9 ± 77.4, and 369.8 ± 45.6 mg·kg^−1^, respectively. There were significant differences among the three groups (*p* < 0.01) and the difference in WSOC content between the D-10 and CK was statistically significant. Similarly, the average WSOC concentrations of CK, D-5, and D-10 at 20–40 cm depth were 168 ± 35.7, 186 ± 24.9, and 313 ± 55.9 mg·kg^−1^, respectively. The difference in WSOC content between the degradation times was statistically significant (*p <* 0.001) and the difference in WSOC content between the D-10 and CK was statistically significant. Regarding MBC content, the average MBC concentrations of CK, D-5, and D-10 at 0–20 cm depth were 497 ± 174, 476 ± 130, and 536 ± 128 mg·kg^−1^, respectively (Table 3). There was no significant difference in MBC content between the degradation times at 0–20 cm depth. However, the average MBC concentrations of CK, D-5, and D-10 at 20–40 cm depth were 334 ± 55.0, 442 ± 57.4, and 443 ± 67.3 mg·kg^−1^, respectively (Table 4). The difference in MBC content between the degradation times was statistically significant (*p <* 0.05). Furthermore, both WSOC and MBC contents in the 0–40 cm soil layer of D-5 and D-10 increased compared to CK. This indicates that degradation time enhances the different C components of the soil C pool.

### 2.2. Changes of Soil Amino Sugars under Different Degradation Times

After treatment of soil samples, glucosamine, galactosamine, epichitosamine and muramic acid were measured. The average glucosamine content of CK, D-5 and D-10 at 0–20 cm was 1780 ± 108, 1980 ± 211 and 2760 ± 193 μg·g^−1^. Degradation time significantly increased the glucosamine content in 0–20 cm soil of Moso bamboo forest, and the difference was statistically significant (*p <* 0.001) (Figure 2). The average glucosamine content of CK, D-5 and D-10 at 20–40 cm was 1870 ± 90.5, 2170 ± 229 and 2480 ± 154 μg·g^−1^. Degradation time significantly increased the glucosamine content in 20–40 cm soil of Moso bamboo forest, and the difference was statistically significant (*p <* 0.001). Among them, D-10 was significantly different from D-5 and CK. The average galactosamine content of CK, D-5 and D-10 at 0–20 cm were 934 ± 118, 1030 ± 111 and 1540 ± 188 μg·g^−1^. Degradation time significantly increased the galactosamine content in 0–20 cm soil of Moso bamboo forest, and the difference was statistically significant (*p <* 0.001) (Table 5). Among them, D-10 was significantly different from D-5 and CK. The average galactosamine content of CK, D-5 and D-10 at 20–40 cm were 945 ± 166, 1170 ± 98.9 and 1490 ± 131 μg·g^−1^. Degradation time significantly increased galactosamine content in 20–40 cm soil of Moso bamboo forest, and the difference was statistically significant (*p <* 0.001). The average epichitosamine content of CK, D-5 and D-10 at 0–20 cm was 0.03 ± 0.01, 0.04 ± 0.02 and 0.07 ± 0.02 μg·g^−1^. Degradation time significantly increased epichitosamine content in 0–20 cm soil of Moso bamboo forest, and the difference was statistically significant (*p <* 0.001). The average epichitosamine content of CK, D-5 and D-10 at 20–40 cm was 0.04 ± 0.01, 0.03 ± 0.01 and 0.06 ± 0.01 μg·g^−1^. Degradation time increased epichitosamine content in 20–40 cm soil of Moso bamboo forest, and the difference was statistically significant (*p <* 0.05). The average muramic acid content of CK, D-5 and D-10 at 0–20 cm were 93.9 ± 22.9, 146 ± 37.7 and 216 ± 31.1 μg·g^−1^. Degradation time significantly increased the muramic acid content in 0–20 cm soil of Moso bamboo forest, and the difference was statistically significant (*p <* 0.001). The average muramic acid content of CK, D-5 and D-10 at 20–40 cm were 111 ± 32.6, 151 ± 33.8 and 186 ± 17.1 μg·g^−1^. Degradation time significantly increased the muramic acid content in 20–40 cm soil of Moso bamboo forest, and the difference was statistically significant (*p <* 0.001). The content of four amino sugars in 0–20 cm soil was higher than that in 20–40 cm soil, which may be because there were more microbial activities in the surface soil of Moso bamboo forest ecosystem, but there was no significant difference between soil depth (Table 6).

The contribution of microbial residues to SOC, referred to as MR/SOC, was investigated in two soil depths: 0–20 cm and 20–40 cm. MR/SOC decreased over time with degradation. Moreover, the highest MR/SOC was observed in the control group (CK), reaching approximately 60%. Statistical analysis revealed a significant difference between FR/SOC and degradation years (*p <* 0.05). However, there was no significant difference observed between MR/SOC and BR/SOC, soil depth, or age of degradation (Figure 3).

### 2.3. Changes of Soil Microbial Residues and Their Contribution to SOC

The mean values of MR, FR, and BR were determined for three degradation treatments (CK, D-5, and D-10) at two soil depths (0–20 cm and 20–40 cm). At 0–20 cm, the MR values were 19.1 ± 1.53, 22.5 ± 2.94, and 31.7 ± 2.58 g·kg^−1^ for CK, D-5, and D-10, respectively. The corresponding BR values were 4.22 ± 1.03, 6.58 ± 1.69, and 9.71 ± 1.39 g·kg^−1^, while the FR values were 14.8 ± 0.82, 16.0 ± 1.55, and 22.0 ± 1.44 g·kg^−1^. At 20–40 cm, the MR values were 20.4 ± 1.59, 24.4 ± 3.11, and 28.3 ± 1.74 g·kg^−1^ for CK, D-5, and D-10, respectively. The corresponding BR values were 4.98 ± 1.46, 6.79 ± 1.52, and 8.38 ± 0.77 g·kg^−1^, while the FR values were 15.44 ± 0.73, 17.58 ± 1.65, and 19.95 ± 1.27 g·kg^−1^ (Table 7 and Table 8).

Statistical analysis showed significant differences in MR, FR, and BR among the degradation treatments (*p <* 0.001), but no significant differences among soil depths. The interaction between MR and FR and degradation age and soil depth was also significant (*p <* 0.05). Specifically, the effect of degradation treatment on MR and FR differed significantly between the two soil depths. These findings suggest that microbial residues play an important role in soil organic matter dynamics and that their response to degradation is influenced by soil depth and time (Table 9). At different soil depths, MR/SOC and FR/SOC decreased with increasing degradation time, but BR/SOC did not change significantly.

MR/SOC is the contribution of microbial residues to SOC. In 0–20 cm and 20–40 cm, MR/SOC decreased with degradation time, and MR/SOC in CK was the highest among the three groups, reaching about 60%. There was statistical difference between FR/SOC and degradation years (*p <* 0.05). There was no significant difference between MR/SOC and BR/SOC and soil depth or age of degradation.

### 2.4. Influences of Soil C Pool on Moso Bamboo Soil Amino Sugars and Soil Microbial Residues

Pearson correlation was employed to investigate the effects of WSOC, MBC, and SOC on soil amino sugars and soil microbial residues. In the 0–20 cm soil layer, there was a significant positive correlation (*p <* 0.01) observed between glucosamine, galactosamine, epichitosamine, and muramic acid and WSOC. Except for epichitosamine, the other three soil amino sugars showed a positive correlation with SOC (*p <* 0.05). However, there was no significant correlation found between the four kinds of soil amino sugars and MBC. Further, there was a significant positive correlation (*p <* 0.05) between MR, BR, and FR and WSOC and SOC, but no significant correlation was found with MBC (Figure 4).

Similarly, in the 20–40 cm soil layer, a positive correlation (*p <* 0.01) was observed between glucosamine, galactosamine, epichitosamine, and muramic acid and WSOC. Glucosamine and muramic acid showed a positive correlation with SOC (*p <* 0.05), while galactosamine and muramic acid were positively correlated with MBC (*p <* 0.05). In addition, MR, BR, and FR soil microbial residues were found to be positively correlated with WSOC (*p <* 0.05) (Figure 4).

## 3. Discussion

### 3.1. Impacts of Degradation Time on Moso Bamboo Soil C Pool and C Components

SOC pools are critical carbon pools for forest ecosystems. Forest management, particularly long-term intensive forest management, has a significant impact on Moso bamboo SOC content [25]. In degraded Moso bamboo forests, SOC reserves increased (Figure 1a,d), which is consistent with Deng et al. [19] and Yuan et al. [16] studies on SOC in degraded Moso bamboo forests. Building on this, we conducted research on soil layers of 0–20 cm and 20–40 cm respectively, and found that SOC in the 0–20 cm layer changed more significantly with degradation time. This could be because the accumulation and rapid decomposition of litter in surface soil increased the SOC content in surface soil. The proportion of old bamboo in Moso bamboo forests without continuous management increases. When old bamboo dies, a large number of old bamboo roots and stems accumulate rapidly in the soil. The decomposition of a large number of bamboo roots also increases SOC reserves. The decomposition of bamboo roots may also be the main reason for the increase of 20–40 cm SOC with degradation. Mancinelli et al. [26] demonstrated that the input of inorganic fertilizer would accelerate the decomposition rate of SOC. Similarly, Sainju et al. [27] showed that irrigation, fertilization, and tillage increased SOC exposure to air and decomposers. These two factors are also responsible for the significant increase in SOC in degraded Moso bamboo forests. Li et al. [18] showed that no fertilization and low-intensity harvesting of Moso bamboo forests also increased soil carbon, which is consistent with our research results.

The degradation time has a significant impact on SOC reserves, as it is closely related to the stability of soil MBC sequestration environment and sufficient substrate. However, the process of degradation leads to the destruction of the above-ground ecosystem of Moso bamboo forest, which could be attributed to the lack of nutrient supply provided by fertilization, as suggested by previous studies [28,29], Degradation further leads to the accumulation of litters in the Moso bamboo forest, and litters are decomposed into the soil, which increases the content of MBC in the soil Further research is needed to investigate the effects of Moso bamboo forest management and degradation to reveal whether C storage and C sequestration efficiency can ultimately be increased in degraded bamboo ecosystems.

WSOC and MBC are two unstable C components of soil that can serve as indicators of soil biological quality [30]. In our study, degradation increased the WSOC content in two soil layers, with a higher increase in the D-10 layer. On one hand, degradation leads to the aging of Moso bamboo forest sample plots, an increase in the number of old bamboo, inadequate nutrient supply for new bamboo growth, and reduced root life activities in the soil. On the other hand, degradation also results in an increased number of dead bamboo, with their roots and stems accumulating on or within the soil surface before eventually decomposing into the soil carbon pool cycle. In addition to the aforementioned decrease in root life activity, it is evident that the supplementation of dead bamboo remains exceeds WSOC consumption significantly; thus, degradation has a significant impact on increasing WSOC levels in Moso bamboo forests [24]. The longer the degradation time, the more accelerated the soil slab, which reduces soil water content [19], leading to weakened soil respiration. Furthermore, the negative correlation between WSOC and water content (Figure 1) could be another reason for the increase in WSOC during degradation. This phenomenon underscores the role of surface litter as a key driver of short-term SOC dynamics. Furthermore, the mention of increased SOC content in deeper soil layers, attributed to the decomposition of bamboo roots, suggests that subsurface SOC dynamics might be influenced by longer-term ecological processes.

Degradation time has no effect on MBC of surface (0–20 cm) soil. The possible despite the absence of exogenously applied substances, the accumulation of litter was observed, leading to a stable MBC in the surface soil in degraded Moso bamboo forest [31]. The insignificant difference in MBC in the surface soil layer could also be explained by the similar hydrothermal conditions caused by litter cover, which is consistent with the research results of Yuan et al. [16]. Conversely, degradation increased the content of MBC in the 20–40 cm soil. This could be related to the decomposition of underground roots, which enhances microbial activity. Studies also proved that soil MBC in bamboo ecosystems was associated with microbial activity [32,33]. WSOC serves as a readily available substrate for microbial decomposition, and its increase might suggest a more labile carbon pool in degraded conditions. This may result in short-term carbon losses if degradation continues unchecked. Meanwhile, MBC, an indicator of microbial biomass and hence microbial activity, showing varying responses across depths suggests that microbial communities might be responding differently to degradation across the soil profile.

In essence, the degradation of Moso bamboo forests, while detrimental in many respects, unveils an intricate play of processes that enrich our understanding of soil carbon dynamics. This knowledge, when harnessed, can aid in devising sustainable forest management practices that not only restore degraded ecosystems but also maximize their potential as carbon sinks.

### 3.2. Response of Soil Amino Sugars and MR to Degradation Time

Degradative processes exert a significant effect on the concentrations of four distinct amino sugars present in the soil matrix. The soil matrix is an intricate, heterogeneous environment, exhibiting a wide range of physical, chemical, and biological properties that can significantly influence the behavior and fate of amino sugars. Understanding these complexities is essential to interpret alterations in amino sugar concentrations. This observed alteration can be ascribed to an augmentation of microbial metabolic activities, culminating in an increased abundance of soil amino sugars. Beyond metabolic activities, the diversity of the microbial community also contributes to variations in amino sugar concentrations. Different microbes preferentially consume or produce certain amino sugars, and the balance of these microbial populations will influence the overall soil amino sugar profile. Amino sugars in soils can either be part of the bioavailable fraction that microbes can readily utilize or adsorbed onto soil particles, rendering them less available. The balance between these fractions can influence overall concentrations. Notably, glucosamine emerges as the most predominant amino sugar constituent, with galactosamine being the second-most predominant contributor. The relative prominence of these compounds underscores their crucial role in the overall compositional framework of soil amino sugars. Soil depth can significantly influence factors like moisture content, temperature, and oxygen availability, which can, in turn, affect microbial activity and the distribution of organic compounds like amino sugars. The amino sugar content in the 0–20 cm soil is higher than that in the 20–40 cm soil. These two conclusions are consistent with the research results of Cui et al. [34] on the temperate forest system in China. The content of amino sugars in the soil is related to the litter content. Degradation leads to litter accumulation and increased microbial activity (MBC can reflect microbial activity to some extent).

We observed that the content of MR was highest in D-10, intermediate in D-5, and lowest in CK (Figure 3a,b). Soil MR is an index representing the time integral of microbial community growth and turnover and is theoretically linked to living microbial biomass [7]. Under similar soil conditions, a higher abundance of microbial biomass and subsequent turnover rate correspond to higher microbial residues [4]. Degradation leads to increased soil microbial abundance and activity, as well as soil MR. This also increases the SOC content of Moso bamboo forest. SOC has a greater affinity for sugars and soils derived from microbial biomass or microbial residues than those derived from plants and is thus more easily stabilized in the soil matrix [35], which has a significant impact on long-term soil C sequestration [36]. We found a positive correlation between the change of MRC. At the same time, MRC, as a part of SOC, accounted for 59.4%, 52% and 34.6% of the total SOC under different degradation times (Figure 5), respectively. It can be seen that with the increase of degradation time, the proportion of MR In SOC gradually decreases, which may be due to the accumulation of bamboo stems and roots of dead bamboo in the soil, while the microbial activity increases, but overall, plant residues may gradually dominate as the degradation time goes on. The contribution degree of MRC to SOC under D-10 treatment is similar to the conclusion of MRC Contribution degree of SOC in forest ecosystem by Wang et al. In our study, the continuously managed plots were disturbed by human activities (such as weeding, regular removal of dead branches and leaves and dead bamboo, etc.), which reduced the transformation of litter and bamboo residues in the soil, resulting in limited transformation of plant residues and subsequent transformation to humus, and the response of microbial activities in the soil to plant residues was relatively slow. At a certain level for a short period of time, the contribution of microbial carbon to soil organic carbon may even occur. In the later period, with the increase of degradation time, the bamboo forest is almost no longer disturbed by human activities, so the litter and bamboo residues covering the soil accumulate in the soil surface, and the process of plant residues and microbial residues in the soil and even their transformation into humus is also enhanced [37]. The most intuitive manifestation is the significant increase of organic carbon in the bamboo forest after 10 years of degradation. In this process, plant residues were the most affected and gradually began to lead the contribution of soil organic carbon, while microbial residues were affected by various influences in the process of transformation to humus. Although the content of microbial residues increased, its growth rate was still unable to compare with the transformation of plant residues, so the contribution to soil organic carbon gradually decreased. In addition, regardless of the degradation time, the contribution of FR to SOC is always higher than that of BR, which is also consistent with the conclusion that FR > BR was found by Wang et al. [38] in the forest. There are two possible reasons for our analysis: 1. Compared with bacteria, fungi can better decompose cellulose, hemicellulose and lignin-containing polymers rich in soil [15]. 2. The breakdown process of fungal cell compounds is slower, and it can also remain in the soil for a longer time FR/SOC is affected by the time of degradation (Table 9), possibly because degraded forests are not fertilized and the decomposition of FR is relatively high, providing a nitrogen source for microorganisms and plants. This is similar to other studies [39,40] in which soil microbial residues were closely related to SOC. SOC is not just a static pool but undergoes constant transformations due to microbial and enzymatic activities. The nature and quantity of the organic inputs, the environmental conditions, and the interactions with soil minerals play crucial roles in determining its composition and stability. Many scholars believe that the strong relationship between the two may be attributed to the following two aspects [4,41,42]. First, SOC is the substrate of soil microbial community activity and thus one of the main determinants of microbial community abundance. Therefore, SOC indirectly affects MR through its influence on microbial community activity. Second, microbial residues are mainly stabilized in the soil through interactions with minerals or metal oxides, and MRC is the primary component of mineral organic C. Li et al. [43] also suggested that nutrient elements such as calcium and magnesium regulate the stability and content of SOC through MR.

The conceptual framework of the soil “Microbial Carbon Pump” (MCP) has been used to demonstrate how microorganisms play an active role in soil C accumulation, and MCP has been proposed as an explanation for C storage processes in soil environments [44]. In this process, soil depth and the soil microbial community have the greatest influence on MCP. Generally, the C content gradually decreases with soil depth, which is also consistent with our research on degraded Moso bamboo forests. Our study on amino sugars and MR in Moso bamboo soil further supports the conclusion of MCP. However, further studies on the specific processes are needed to fully understand the relationship between soil depth, microbial communities, and C accumulation in soil.

### 3.3. Application of Microbial Residual C Research in Moso Bamboo Forest

Moso bamboo is distinguished for its robust carbon sequestration capabilities. In China, the carbon storage within the soil of Moso bamboo forests constitutes approximately 75% of the total carbon storage within the bamboo ecosystem [45]. The lifecycle, structural attributes, microbial activities, and response mechanisms of soil microorganisms are integral components of the soil’s biogeochemical cycle. These factors significantly influence the formation and turnover of soil organic matter. Despite its importance, there exists a notable dearth of research concerning MR in Moso bamboo forests. Moreover, the carbon conversion efficacies of soil fungi and bacteria within these forests remain inadequately elucidated. In our study, we endeavored to elucidate the role and contribution of soil fungi and bacteria in degraded Moso bamboo forests. However, measuring soil amino sugars, which are imperative for assessing the microorganisms’ contribution to SOC, poses intricate challenges. We analyzed the contribution of soil fungi and bacteria to degraded Moso bamboo forests (Figure 4). However, compared with SOC, MBC, and WSOC, measuring soil amino sugars is more challenging and complicated, making it. Difficult to assess the contribution of microorganisms to SOC in the degraded Moso bamboo forest ecosystem. To overcome this challenge, we conducted a linear analysis using SPSS to construct a linear prediction equation (Table 10) for MR of different soil layers in the degraded Moso bamboo forest ecosystem. This method allowed us to estimate the MR of Moso bamboo forests, and better understand the contribution of MR to SOC and its influence on the C cycle of bamboo soil. Our findings are valuable for Moso bamboo forest management and bamboo soil C cycle research.

### 3.4. Limitations

In our study, we also explained that soil temperature, moisture content, bulk density, pH and other properties will also have an impact on MR. In this study, samples were only taken from subtropical bamboo forests, and the differences in soil temperature, moisture content, bulk density, pH and other properties were not very significant. Further experiments in different ecosystems and soils with different compositions are needed to explore changes in the contribution of MR to SOC at different temperatures, water content, bulk density or pH, or even in different ecosystems [38].

Although the influence of litter was removed when the soil samples were collected in this study, humus still existed in the topsoil, especially in the degraded bamboo forest, and the humus layer had an impact on the flora, fauna and microbial residues in the soil. The influence of humus on microbial residues and organic matter should be taken into more careful consideration in the subsequent research. At the same time, some scholars believe that in forest ecosystems, in fact, the contribution of plant residues to SOC is relatively large, and in subsequent studies, the contribution of plant residues and microbial residues in soil to SOC should be comprehensively compared to get a deeper response mechanism.

Microorganisms also play an important role in the transformation process of soil organic matter [46]. Microorganisms decompose SOM through mineralization and release CO_2_ in the atmosphere, and at the same time convert SOM into their own components through assimilation [44]. Therefore, the role of microbial residues in the accumulation and transformation process of SOM has attracted more and more attention [47,48]. Zhang Wei explored the influence of microbial residues on the formation of soil humus under pure culture conditions, and found that humus could be extracted after the death and fragmentation of microbial cells, but with the extension of culture time, the amount of extraction showed a trend of decline. The distribution of microbial biomass carbon in each component of humus was mainly in fulvic acid (FA). However, under liquid culture conditions, no resynthesis of humus has been found, which indicates that humus will eventually be mineralized into CO_2_ and water in the absence of microbial available carbon source input [49]. All the above researches provide a theoretical basis for us to further explore the transformation relationship between microbial residue carbon and humus. 

## 4. Materials and Methods

### 4.1. Description of Study Area

The study area is selected in An’ji County, Huzhou City, Zhejiang Province, between 119°14′~119°53′ east longitude and 30°23′~30°53′ north latitude (Figure 6). Anji County is located in the Yangtze River Delta region. Anji County belongs to the north subtropical monsoon climate zone, sufficient light, abundant precipitation, mild climate, four distinct seasons, topography relief height difference, vertical climate obvious. Anji County is the birthplace of the idea that “Lucid waters and lush mountains are invaluable assets”. The area of Moso bamboo forest in Anji County is 665.33 km^2^, accounting for 49.25% of the forest area respectively. Anji County is the hometown of bamboo in China. It ranks first in China in five indicators: bamboo quantity, annual output of bamboo, annual output value of bamboo industry, annual export value of bamboo products and comprehensive economic strength of bamboo industry. The main soil types in Zhejiang Province are red soil and yellow soil [50,51]. The soil of Moso bamboo forest in Anji County, Zhejiang Province was studied under different degradation of subtropical forest. Specifically, the density of Moso bamboo forests before degradation (continuous management) was 2400–2700 trees per hectare(Table 11). (We obtained permission from Anji County Forestry Bureau to sample Moso bamboo. The bamboo species are classified and identified by Anji County Forestry Bureau according to the requirements of China. Xuekun Cheng and Yongjun Shi also identified bamboo samples in the lab).

### 4.2. Experiment Design

In order to investigate the difference of soil C content and microbial residue characteristics of bamboo forest ecosystem under different degradation years, the spatio-temporal substitution method was used in this study. Specifically, we selected Moso bamboo forest areas with varying degradation years in Anji County, Zhejiang province and divided them into three groups: normal continuous management (CK), 5-year degradation (D-5) and 10-year degradation (D-10), based on the management history records and actual operation conditions of Anji County Forestry Bureau. To ensure consistency in stand conditions, including terrain and soil, and to ensure that all pure Moso bamboo stands were under continuous operation before degradation, each sample point was selected based on the following two criteria. First, stand conditions were relatively consistent, and second, all pure Moso bamboo stands were under continuous operation before degradation. While CK implemented regular bamboo shoots harvesting, weeding, fertilization, and tillage, the other groups only underwent bamboo shoots harvesting. To mitigate environmental biases and enhance the validity of our study, a randomized block design is used for the experimental framework. For precision, we designated 5 plots within each category, culminating in a comprehensive set of 15 standardized plots. Each fixed plot measured 10 m on one side, with an area of 100 square meters. To collect soil samples, we set up five soil sample sites in the southeast, southwest, northeast, northwest, and central part of the fixed plot. We dug up to 40 cm for each soil profile, and soil samples were collected from 0–20 cm and 20–40 cm layers. Soil from the 5 sites was sent back to the lab.in order to measure SOC, MBC, WSOC and amino sugars.

In August 2022, soil samples were collected from the study sites. Prior to sampling, the surface litter layer was removed, and soil was sampled at depths of 0–20 cm and 20–40 cm. The collected samples were stored in an ultra-low temperature refrigerator set at −40 °C. After the removal of litter and residual roots, the samples were divided into two parts using a 2 mm sieve. One part was stored at −4 °C for the determination of MBC and WSOC, while the other part was air-dried and screened using a 0.15 mm screen for the measurement of soil amino sugars and other analyses at 25 °C. Soil pH was measured using a pH meter on soil and water suspensions (1:2.5 *w*/*v*). Soil moisture content (Ms, %) was determined by drying samples in a 105 °C oven for 24 h, following the Agricultural Chemistry Committee of China (2000) protocol. SOC in soil samples was measured using the potassium dichromate external heating method, as described by Zhang et al. [52] and Nóbrega et al. [53]. The chloroform fumigation extraction method, as described by Vance et al. [54], was used to determine soil microbial biomass carbon (MBC, mg kg^−1^). Soil water-soluble organic carbon (WSOC, mg kg^−1^) was measured using the method by Singh et al. [55]. The C and N concentrations were analyzed using the TOC-TN automatic analyzer (TOC-Vcph, Shimadzu, Kyoto, Japan).

### 4.3. Determination of Soil Microbial Residues

Soil amino sugars were hydrolyzed with hydrochloric acid and determined by gas chromatography. The method of Zhang and Amelung [56] was used for determination. Agilent 7890B meteorological chromatograph (FID hydrogen flame ionization detector) was used for detection.

Microbial residues were measured by soil amino sugars using the following Equation [11,13,14].
(1)Fungal residue C=(mmol glucosamine−2×mmol muramic acid)×179.2×9
(2)Bacterial residue C=muramic acid×45
(3)Microbial residue C=Fungal residue C+Bacterial residue C
(4)Contribution of MR to SOC=MR/SOC
(5)Contribution of FR to SOC=FR/SOC
(6)Contribution of BR to SOC=BR/SOC
where 179.2 is the molecular mass of glucosamine, while 9 and 45 are conversion factors for the conversion of fungal glucosamine to fungal residual C and bacterial cytosolic acid to bacterial residual C, respectively.

### 4.4. Statistical Analysis

All data were analyzed by using SPSS 26.0 (SPSS Inc., Chicago, IL, USA) with a significance level of 0.05. One-way analysis of variance (ANOVA) and minimum significant difference (LSD) tests were used to investigate the effects of degradation time on soil C components and microbial residues. Repeated measurement analysis of variance (RM-variance) and LSD tests were used to analyze MBC and WSOC. All data were tested for variance uniformity and distribution normality by S-W test in SPSS before variance analysis. In addition, we used Pearson correlation analysis to explore the correlation between soil microbial residues and soil C pool (SOC, WSOC and MBC) in Moso bamboo forest under different degradation years.

## 5. Conclusions

In this study, the effects of different degradation years on SOC content, MRC content, and soil amino sugars in Moso bamboo forest were investigated. The study found that SOC, WSOC and MBC contents increased with degradation, confirming the hypothesis (1). The effect of degradation on MR content of Moso bamboo forest was found to depend on the length of degradation time, consistent with the hypothesis (2). The change in MR content in Moso bamboo forest was found to be correlated with the change in soil C pool, but there was no significant correlation between MBC at 0–20 cm and MR change, confirming hypothesis (3). The study established a linear equation between soil C and MR, making it easier to predict MR levels in Moso bamboo forests. The results of this study have great significance for Moso bamboo forest management. Since SOC plays a key role in determining MR, management strategies that can improve SOC levels should be implemented. The study suggests that the input of organic matter or litter in Moso bamboo forest ecosystems can effectively stimulate MR levels, which is of great significance for the long-term C storage of SOC.

This study is the first to apply MR method to Moso bamboo forest, and expanding the study of soil microorganisms and MR in Moso bamboo forest is crucial for understanding the response mechanism of soil microorganisms to different environments and the C sequestration process of MR in Moso bamboo forest. Furthermore, this study deepens our understanding of the impact of MR reserves on degraded subtropical Moso bamboo forests and provides basic data support for soil C sequestration for the restoration and management of bamboo plantations. The study’s findings will also help to predict soil C storage in Moso bamboo forests more accurately.

The results obtained in this study deepen our understanding of the impact of MR reserves on subtropical degraded Moso bamboo forests and provide essential data support for soil carbon sequestration in Moso bamboo forest restoration and management. Additionally, these findings will help to more accurately predict soil carbon stocks in Moso bamboo forests.

## Figures and Tables

**Figure 1 plants-13-01526-f001:**
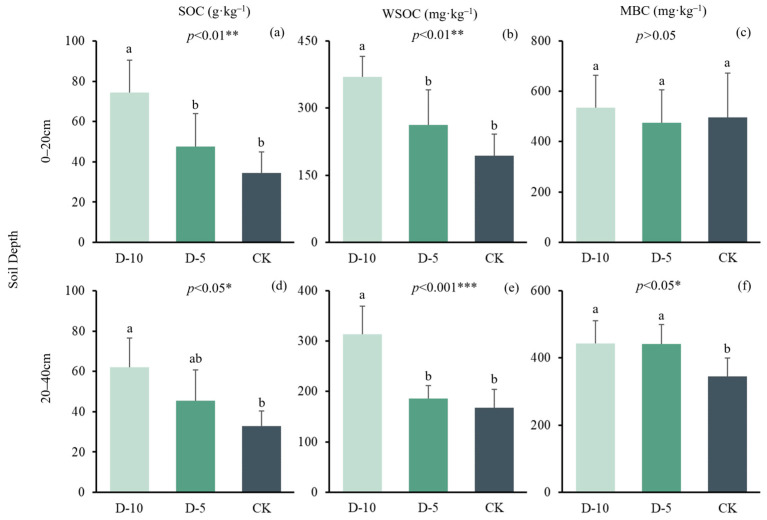
The content of soil organic carbon (SOC), microbial carbon (MBC) and water-soluble carbon (WSOC) in CK, D-5 and D-10. *p* represents a significant difference between groups. *, ** and *** respectively represent *p <* 0.05, *p <* 0.01 and *p <* 0.001. The letters represent differences between different degradation time in the same group. (**a**–**c**) are WSOC, MBC and SOC Pearson correlation analysis at 0–20 cm soil depth, (**d**–**f**) are WSOC, MBC and SOC Pearson correlation analysis at 20–40 cm soil depth.

**Figure 2 plants-13-01526-f002:**
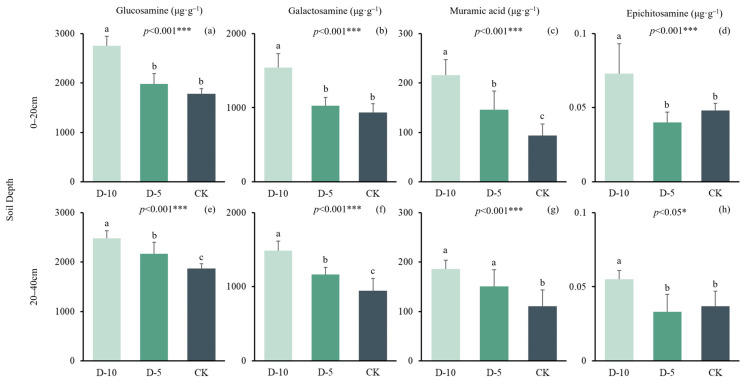
The content of glucosamine, galactosamine, muramic acid and epichitosamine in CK, D-5 and D-10. *p* represents a significant difference between groups. *, and *** respectively represent *p <* 0.05 and *p <* 0.001. The letters represent differences between different degradation time in the same group. (**a**–**d**) are WSOC, MBC and SOC Pearson correlation analysis at 0–20 cm soil depth, (**e**–**h**) are WSOC, MBC and SOC Pearson correlation analysis at 20–40 cm soil depth.

**Figure 3 plants-13-01526-f003:**
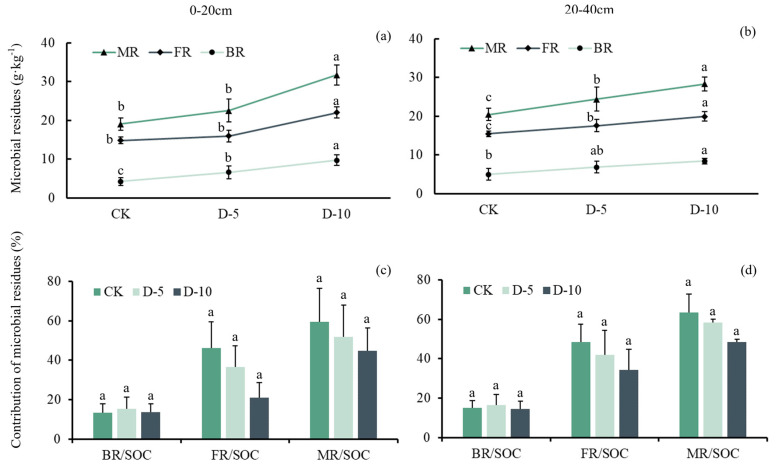
Changes in the influences of degradation on microbial residues (**a**,**b**) and the contribution of microbial residues to Moso bamboo soil (**c**,**d**). MR means microbial residues; FR means fungal residues; BR means bacterial residues. MR/SOC, FR/SOC and BR/SOC mean their contribution to soil organic C, and show their contribution to soil organic C. The letters represent differences between different degradation time in the same group.

**Figure 4 plants-13-01526-f004:**
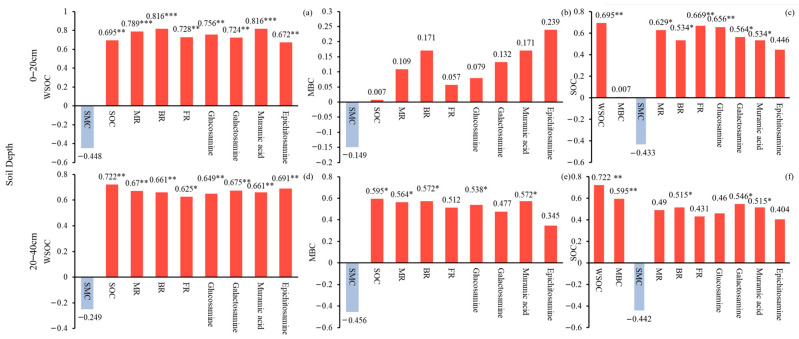
Pearson correlation analysis of soil carbon and microbial residues. *, ** and *** respectively represent *p <* 0.05, *p <* 0.01 and *p <* 0.001, represent the correlation significance. (**a**–**c**) are WSOC, MBC and SOC Pearson correlation analysis at 0–20 cm soil depth, (**d**–**f**) are WSOC, MBC and SOC Pearson correlation analysis at 20–40 cm soil depth.

**Figure 5 plants-13-01526-f005:**
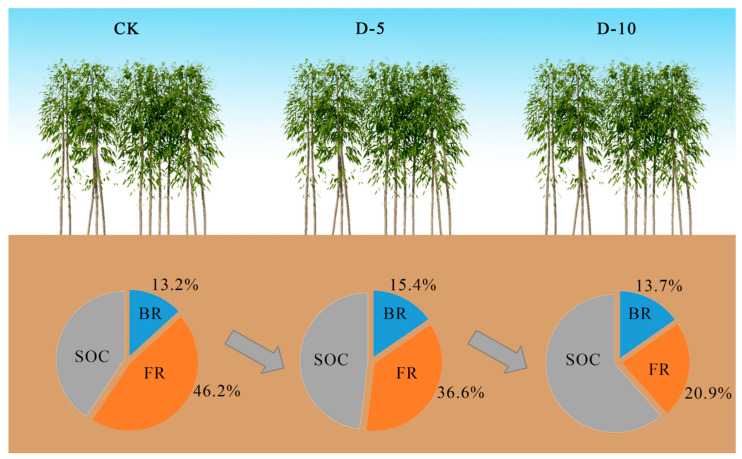
Effect of degradation time on contribution of microbial residues to SOC (the whole circle represents the SOC).

**Figure 6 plants-13-01526-f006:**
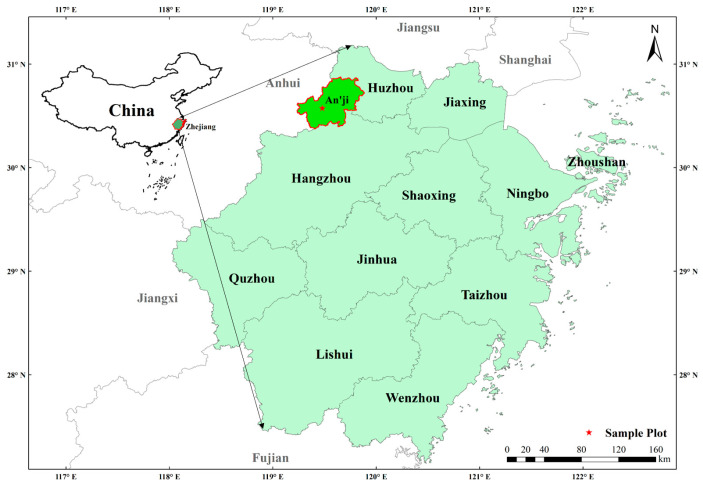
Location of the studied Moso bamboo forest. The base image is from National Geographic Information Public Service Platform (https://www.tianditu.gov.cn (accessed on 12 April 2023)). Image processing using ArcMap (10.8).

**Table 1 plants-13-01526-t001:** Basic soil environmental factors at 0–20 cm depth.

Soil Depth	Treatment	SBD	pH	SMC	df	N
		(g·cm^−3^)		(% *w*/*w*)		
0–20 cm	D-10	1.26 ± 0.06 a	5.24 ± 0.34 a	18.6 ± 5.72 b	2	15
	D-5	1.30 ± 0.05 a	5.32 ± 0.33 a	17.2 ± 2.38 b	2	15
	CK	1.26 ± 0.04 a	5.44 ± 0.30 a	25.9 ± 3.85 a	2	15

Note: Different lowercase letters in each column indicate the differences of soil physicochemical indexes at different degradation times at *p* < 0.05 significance level under the condition of minimum significance difference (LSD) test.

**Table 2 plants-13-01526-t002:** Basic soil environmental factors at 20–40 cm depth.

Soil Depth	Treatment	SBD	pH	SMC	df	N
		(g·cm^−3^)		(% *w*/*w*)		
20–40 cm	D-10	1.57 ± 0.02 a	5.11 ± 0.17 a	17.0 ± 4.80 a	2	15
	D-5	1.58 ± 0.06 a	5.11 ± 0.19 a	17.7 ± 2.37 a	2	15
	CK	1.54 ± 0.07 a	5.12 ± 0.06 a	24.2 ± 7.14 a	2	15

Note: The SBD means soil bulk density. The SMC means soil moisture content. Average ± standard deviation within differences between different degradation times and soil pH, bulk density and moisture content. Different lowercase letters in each column indicate the differences of soil physicochemical indexes at different degradation times at *p* < 0.05 significance level under the condition of minimum significance difference (LSD) test.

**Table 3 plants-13-01526-t003:** SOC, WSOC and MBC at 0–20 cm depth.

Soil Depth	Treatment	SOC	WSOC	MBC	df	N
		(g·kg^−1^)	(mg·kg^−1^)	(mg·kg^−1^)		
0–20 cm	D-10	74.3 ± 8.07 a	370 ± 45.6 a	536 ± 128 a	2	15
	D-5	47.5 ± 8.24 b	263 ± 77.4 b	476 ± 130 a	2	15
	CK	34.4 ± 5.29 b	193 ± 49.0 b	497 ± 174 a	2	15

Note: Different lowercase letters in each column indicate the differences of soil physicochemical indexes at different degradation times at *p* < 0.05 significance level under the condition of minimum significance difference (LSD) test.

**Table 4 plants-13-01526-t004:** SOC, WSOC and MBC at 20–40 cm depth.

Soil Depth	Treatment	SOC	WSOC	MBC	df	N
		(g·kg^−1^)	(mg·kg^−1^)	(mg·kg^−1^)		
20–40 cm	D-10	62.0 ± 7.30 a	313 ± 55.9 a	443 ± 67.3 a	2	15
	D-5	45.5 ± 7.61 ab	186 ± 24.9 b	442 ± 57.4 a	2	15
	CK	39.0 ± 5.59 b	168 ± 35.7 b	334 ± 55.0 b	2	15

Note: Different lowercase letters in each column indicate the differences of soil physicochemical indexes at different degradation times at *p* < 0.05 significance level under the condition of minimum significance difference (LSD) test.

**Table 5 plants-13-01526-t005:** Glucosamine, galactosamine, muramic acid and epichitosamine at 0–20 cm depth.

Soil Depth	Treatment	Glucosamine	Galactosamine	Muramic Acid	Epichitosamine	df	N
		(μg·g^−1^)	(μg·g^−1^)	(μg·g^−1^)	(μg·g^−1^)		
0–20 cm	D-10	2760 ± 193 a	1540 ± 188 a	216 ± 13.9 a	0.07 ± 0.02 a	2	15
	D-5	1980 ± 211 b	1030 ± 111 b	146 ± 37.8 b	0.04 ± 0.02 b	2	15
	CK	1780 ± 108 b	934 ± 118 b	93.9 ± 22.9 c	0.03 ± 0.01 b	2	15

Note: Different lowercase letters in each column indicate the differences of soil physicochemical indexes at different degradation times at *p* < 0.05 significance level under the condition of minimum significance difference (LSD) test.

**Table 6 plants-13-01526-t006:** Glucosamine, galactosamine, muramic acid and epichitosamine at 20–40 cm depth.

Soil Depth	Treatment	Glucosamine	Galactosamine	Muramic Acid	Epichitosamine	df	N
		(μg·g^−1^)	(μg·g^−1^)	(μg·g^−1^)	(μg·g^−1^)		
20–40 cm	D-10	2480 ± 154 a	1490 ± 131 a	186 ± 7.66 a	0.06 ± 0.01 a	2	15
	D-5	2170 ± 229 b	1170 ± 98.9 b	151 ± 15.1 a	0.03 ± 0.01 b	2	15
	CK	1870 ± 90.5 c	945 ± 166 c	111 ± 14.6 b	0.04 ± 0.01 b	2	15

Note: Different lowercase letters in each column indicate the differences of soil physicochemical indexes at different degradation times at *p* < 0.05 significance level under the condition of minimum significance difference (LSD) test.

**Table 7 plants-13-01526-t007:** Changes in the influences of degradation on microbial residues at 0–20 cm depth.

Soil Depth	Treatment	MR	BR	FR	MR/SOC	FR/SOC	BR/SOC	df	N
		(g·kg^−1^)	(g·kg^−1^)	(g·kg^−1^)	(%, *w*/*w*)	(%, *w*/*w*)	(%, *w*/*w*)		
0–20 cm	D-10	31.7 ± 2.58 a	9.71 ± 1.39 a	22.0 ± 1.44 a	44.6 ± 5.20 a	46.2 ± 5.87 a	13.7 ± 1.87 a	2	15
	D-5	22.5 ± 2.94 b	6.58 ± 1.69 b	16.0 ± 1.55 b	51.9 ± 7.20 a	36.6 ± 4.82 a	15.4 ± 2.59 a	2	15
	CK	19.1 ± 1.53 c	4.22 ± 1.03 c	14.8 ± 0.82 b	59.3 ± 7.61 a	30.9 ± 3.39 a	13.2 ± 2.00 a	2	15

Note: Different lowercase letters in each column indicate the differences of soil physicochemical indexes at different degradation times at *p* < 0.05 significance level under the condition of minimum significance difference (LSD) test.

**Table 8 plants-13-01526-t008:** Changes in the influences of degradation on microbial residues at 20–40 cm depth.

Soil Depth	Treatment	MR	BR	FR	MR/SOC	FR/SOC	BR/SOC	df	N
		(g·kg^−1^)	(g·kg^−1^)	(g·kg^−1^)	(%, *w*/*w*)	(%, *w*/*w*)	(%, *w*/*w*)		
20–40 cm	D-10	28.3 ± 1.74 a	8.38 ± 0.77 a	20.0 ± 0.33 a	48.4 ± 6.49 a	34.2 ± 4.06 a	14.3 ± 1.84 a	2	15
	D-5	24.4 ± 3.11 b	6.79 ± 1.52 a	17.6 ± 0.74 b	58.3 ± 7.87 a	42.0 ± 5.56 a	16.3 ± 2.49 a	2	15
	CK	20.4 ± 1.59 c	4.98 ± 1.46 b	15.4 ± 0.57 c	63.5 ± 4.17 a	48.4 ± 4.70 a	15.2 ± 1.57 a	2	15

Note: Different lowercase letters in each column indicate the differences of soil physicochemical indexes at different degradation times at *p* < 0.05 significance level under the condition of minimum significance difference (LSD) test.

**Table 9 plants-13-01526-t009:** Influences of degradation years on microbial residues and their contribution to the SOC.

	Soil Depth	Degradation Years	Degradation Years × Soil Depth
SOC	0.30	<0.001	0.61
MR	0.94	<0.001	<0.05
FR	0.91	<0.001	<0.05
BR	0.81	<0.001	0.22
MR/SOC	0.38	0.09	0.97
FR/SOC	0.36	<0.05	0.94
BR/SOC	0.61	0.63	0.93

Note: *p <* 0.05 indicated statistical significance.MR means microbial residues; FR means fungal residues; BR means bacterial residues. MR/SOC, FR/SOC and BR/ SOC mean Their contribution to soil organic carbon, and show their contribution to soil organic carbon. × is the interaction between the degradation years and soil depth.

**Table 10 plants-13-01526-t010:** The linear regression models of FR and BR at different soil depths.

	Soil Depth	Linear Regression Model	R^2^	*p*
FR	0–20 cm	y = 0.043 + 0.116SOC − 0.114MBC + 0.360WSOC	0.60	<0.05 *
FR	20–40 cm	y = 0.028 − 0.204SOC + 0.452MBC + 0.326WSOC	0.63	<0.01 **
BR	0–20 cm	y = 0.016 − 0.128SOC − 0.236MBC + 1.120WSOC	0.69	<0.01 **
BR	20–40 cm	y = −0.021 − 0.358SOC + 0.995MBC + 0.678WSOC	0.68	<0.01 **

Note: The data were normalized and linear regression models were established. R^2^ represents the degree of fit of the curvilinear regression. The VIF (Severity of multicollinearity) of all models are <5. *p* represents the significance of F test of each model, * and ** respectively represent *p <* 0.05 and *p <* 0.01, represent the correlation significance.

**Table 11 plants-13-01526-t011:** Site baseline data and information.

Plot	Elevation	Slope	Living Bamboo Density	D.B.H
	(m)		(Trees ha^−1^)	(cm)
CK-1	103	9	2470	11.7
CK-2	115	12	2496	12.5
CK-3	151	13	2560	11.9
CK-4	130	17	2673	13.2
CK-5	97	11	2614	12.8
D-5-1	76	21	1772	12.4
D-5-2	143	18	1742	12.1
D-5-3	135	20	1802	11.9
D-5-4	126	21	1792	11.5
D-5-5	112	17	1748	11.7
D-10-1	140	10	1341	12.9
D-10-2	173	5	1349	13.5
D-10-3	92	12	1390	13.1
D-10-4	86	14	1427	12.8
D-10-5	166	28	1301	11.9

## Data Availability

The datasets analyzed during this study are not publicly available due to legal restrictions, but are available from the corresponding author on a reasonable request.

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
