# Peer review of "Responses of Soil Carbon and Microbial Residues to Degradation in Moso Bamboo Forest"

_plants, 2024, doi:10.3390/plants13111526_

Round 1

Reviewer 1 Report

Comments and Suggestions for Authors

This study investigated the effects of different degradation durations (continuous management, 5 years of degradation, and 10 years of degradation) on soil carbon and microbial residues in Moso bamboo forests. it provides insights into the effects of degradation on soil carbon dynamics and microbial residue contributions in Moso bamboo forests, emphasizing the potential role of microbial residues in carbon sequestration and sustainable forest management. Although this manuscript presents some interesting findings, significant revisions are necessary before considering this manuscript for publication.

1.  In the introduction, it was mentioned that this study is built on the authors' previous findings. They need to provide a better rationale for studying soil carbon and microbial residues in degraded Moso bamboo forests by addressing the important knowledge gap in this field of soil C dynamics. 

2. This study describes the changes in soil carbon pools, amino sugars, and microbial residues under different degradation durations, but it does not provide clear mechanistic explanations for these observed changes. The discussion section primarily focuses on stating the findings but lacks in-depth analysis of the underlying processes and factors driving the observed patterns. Accordingly, more in-depth discussions are required. 

3. While the results showed differences in soil properties and microbial residues between the 0-20 cm and 20-40 cm soil layers, it is unclear whether degradation may impact soil carbon dynamics and microbial communities at deeper soil depths. Thus, please consider to explore depth-related patterns to yield better insights into soil C dynamics.

4. This study employed a space-for-time substitution approach, which assumes that the sites were initially similar before the onset of degradation. The authors need to provide the baseline data or information on the initial conditions of the sites.

5. There are some issues with the significant digits reported for the data. Please follow standardized guidelines for reporting significant digits in scientific publications and report measured values with an appropriate number of significant digits based on the precision of the measurement methods used.

6. The presentation of statistical results is limited to reporting the p-values (e.g., p < 0.05, p < 0.001). It is generally recommended to report the full test along with the degrees of freedom and sample sizes. This information is crucial for interpreting the strength and magnitude of the observed effects.

7. Pearson correlation analysis was used to investigate the relationships between soil carbon pools, amino sugars, and microbial residues. However, this approach does not account for the inter-correlation among SOC, MBC and WSOC. Meanwhile, the equations obtained by this approach requires validation with independent data sets to increase their reliability.

Comments on the Quality of English Language

Minor editing of English language is required. 

Reviewer 2 Report

Comments and Suggestions for Authors

This informative paper needs significant editing. Suggest moving Section 5 to before current Section 2.  Delete extra verbage such as "We discerned", "Results indicate that", "We observed that", "We found that", "The study found that", "The study suggests that".  "wool bamboo"?  Some inconsistent format in References.  Change format for citations?  Some typos.  Add significance indicator to Table 1.  Define a-f in Figure 4 caption.  Is Figure 5 cited?

Comments on the Quality of English Language

Much editing needed.

Reviewer 3 Report

Comments and Suggestions for Authors

This is an interesting but strongly below standard ms. Degradation is not explained in Mat and Meth.

The statement that MRC is the greatest C pool in soil is not supported by mst of the results presented by a great number of scientific contributions.

The authors suggest that amino sugars are very stable in soils. Compared to which other classes of organic soil compounds? What about the stability of lignin its degradation and reaction products such as humic substances? Why the focus on amino sugars in this ms?

And what is regenerative agriculture considering the lack of a definition of it?

This paper requires major revision, at least.

Comments on the Quality of English Language

no specific!

Round 2

Reviewer 1 Report

Comments and Suggestions for Authors

Please ensure that reported data adheres to significant digit conventions. For instance, if the data is 1234.56 ± 110.00, it should be rounded to 1230 ± 110.

Comments on the Quality of English Language

Please review this manuscript with a focus on identifying and correcting typos and grammatical errors.

Author Response

First of all, we would like to thank you for your help with our ms. We have corrected the data presentation according to the significant digit conventions you provided, and in response to the typos and grammatical errors problem you raised, we have also started to correct and change, thank you again for your comments!

Reviewer 3 Report

Comments and Suggestions for Authors

see the comments below.

Comments on the Quality of English Language

I cannot see any substantial revisions by the authors.. The statement that more than 50% of SOC may come from MRC in soil should be treated with caution. SOC mainly consits of chemically, biochemcally and sometimes biological transformed organic matter from various origin including dead microbial carbon. The paper of Lehmann and Kleber, 2015, Nature,  has, among others introduced some misunderstanding. See the comments among others by Gerke, 2018, Agronomy, 8, Hayes and Swift, 2020, Adv. Agron., Bloom et al., 2019, J. Environ. Qual., Piccolo, 2016, Chem. Biol. Technol. Agric., Duo et al., 2020, Pedoshere. Unless the authors give no balanced overview on SOC in the intro and a balanced discussion of their results, the paper should not be accepted. 

Author Response

Thank you again for your comments on ms. We have carefully read the several references you mentioned and have confirmed and revised the ideas in our manuscript in conjunction with our references. Admittedly, the statement that "more than 50% of SOC in soil may come from MRC" should be considered with caution and we have adjusted it to a more reasonable statement. At the same time, when we wrote the manuscript, we found that some scholars comprehensively considered the determination of lignin and amino sugar indexes, and explained the accumulation of soil organic carbon from both plant residues and microbial residues. However, quite a few scholars only focused on the study of the influence of microbial residues carbon on soil organic carbon accumulation, and reached a reasonable conclusion. We also comprehensively took these conclusions into account in the analysis and comparison of the test results in the manuscript, and finally got the response mechanism of the microbial residues and even soil organic carbon in the bamboo forest under different degradation times. It is true that we have not been able to further explore humus in the study of SOC in Moso bamboo forests, which will be further taken into account in future studies.(4.4 Limitations) In our ms, we refer to the following paper: Wang et al., 2021, Soil Biology and Biochemistry, 162:108422; Hu et al.,2020, Soil Biology and Biochemistry,141:107660; Ding et al.,2019, Soil Biology and Biochemistry,135:13-19; Shao et al.,2017, Soil Biology and Biochemistry,114:114-120; Liang et al.,2019, Global Change Biology, 25:3578–3590. All the above papers separately explored the accumulation or contribution of microbial residues to soil organic carbon. The above papers also indicate that our research direction on the contribution of MR To SOC in the soil of bamboo forest is feasible.

In conclusion, we would like to include the lignin and humus issues mentioned in your comments in the limitations of our manuscript research, and I would be grateful if you would accept.

Round 3

Reviewer 3 Report

Comments and Suggestions for Authors

The authors mostly ignored the critics. The references cited do not take into account the interactions of microbia l residues further reacting chemically, biochemically and forming what is called humic matter or humic substances. The main question is whether unchnged microbial residues or necromass will persist unreacted i.e. unstabilized in soil accounting for more than 50% of Corg.

Comments on the Quality of English Language

see above

Author Response

Thank you again for your suggestions on our manuscript and your help in revising it. To be honest, due to funding and research priorities, we did not include humus in the indicators to be measured in the design of this study, which resulted in ignoring the transformation relationship between microbial residues and humus in soil. We believe that your criticism is reasonable. However, our current data do not support a more in-depth analysis of the transformation process between microbial residues and humus, please forgive me.

With regard to your question "The main question is whether unchnged microbial residues or necromass will persist unreacted i.e. unstabilized in soil accounting for more than 50% of Corg. " We have also confirmed the experimental data, and in the treatment of CK and D-5, the contribution ratio of MRC is indeed higher than that of forest soil in the literature we referred to. We think there may be differences between human activities and business models. Just as I revised in this manuscript, the Phyllostachys CK group underwent a large number of human disturbances (such as timely cleaning of litter and Moso bamboo). This series of human disturbances may lead to a decrease in the conversion rate of plant residues in the soil in a short period of time, which may also affect the conversion process of humus. However, after ten years of degradation, the Moso bamboo forest is basically in the stage of abandoning management, and there is basically no human interference. Litters and Moso bamboo residues accumulate on the soil, which may lead to a sharp increase in vegetation residues in the soil and accelerate the transformation process of vegetation residues into humus. Finally, the accumulation of humus also leads to an increase in soil organic carbon, which is also consistent with our test results. We try to explain the main question you raised from the above aspects, which is also what we can analyze or discuss in the experimental data. We hope that through our explanation, you can change your mind about this main question (in fact, the contribution of MRC in SOC over 50% is indeed too high. However, with the passage of degradation time, the contribution of carbon from plant residues will most likely surpass that of MRC, and just as you said, MRC will also transform into humus in this process, but we really cannot give a more detailed description of the specific transformation degree in this paper. In future soil studies, we will incorporate humus indexes into the experimental design for further description.)

All in all, the experimental data of humus cannot be supplemented in this paper due to the aging of soil samples and other reasons. However, after further consulting the papers of other scholars, we have included their relevant research content on microbial residues and humus in the literature review and subsequent analysis. We would be grateful if you could approve the manuscript.

I am sending you this study on microbial residues related to humus as a PDF, which I think can be used as part of the analysis.

Round 4

Reviewer 3 Report

Comments and Suggestions for Authors

Most critics were answered by the authors and some corrections were made. The paper is not fully satisfying but should be published now.

Comments on the Quality of English Language

see above